# The COX-2/PGE$_2$ pathway suppresses apical elimination of RasV12-transformed cells from epithelia

Nanami Sato[1], Yuta Yako[1], Takeshi Maruyama[1], Susumu Ishikawa[1], Keisuke Kuromiya[1], Suzumi M. Tokuoka[2], Yoshihiro Kita[2,3] & Yasuyuki Fujita[1✉]

At the initial stage of carcinogenesis, when RasV12-transformed cells are surrounded by normal epithelial cells, RasV12 cells are apically extruded from epithelia through cell competition with the surrounding normal cells. In this study, we demonstrate that expression of cyclooxygenase (COX)−2 is upregulated in normal cells surrounding RasV12-transformed cells. Addition of COX inhibitor or COX-2-knockout promotes apical extrusion of RasV12 cells. Furthermore, production of Prostaglandin (PG) E$_2$, a downstream prostanoid of COX-2, is elevated in normal cells surrounding RasV12 cells, and addition of PGE$_2$ suppresses apical extrusion of RasV12 cells. In a cell competition mouse model, expression of COX-2 is elevated in pancreatic epithelia harbouring RasV12-exressing cells, and the COX inhibitor ibuprofen promotes apical extrusion of RasV12 cells. Moreover, caerulein-induced chronic inflammation substantially suppresses apical elimination of RasV12 cells. These results indicate that intrinsically or extrinsically mediated inflammation can promote tumour initiation by diminishing cell competition between normal and transformed cells.

[1] Division of Molecular Oncology, Institute for Genetic Medicine, Hokkaido University Graduate School of Chemical Sciences and Engineering, Sapporo, Hokkaido 060-0815, Japan. [2] Department of Lipidomics, Graduate School of Medicine, The University of Tokyo, Tokyo 113-0033, Japan. [3] Life Sciences Core Facility, Graduate School of Medicine, The University of Tokyo, Tokyo 113-0033, Japan. ✉email: yasu@igm.hokudai.ac.jp

At the initial stage of carcinogenesis, an oncogenic mutation occurs in single cells within the epithelial layer. The newly emerging transformed cells and the surrounding normal epithelial cells often compete with each other for survival; this phenomenon is called cell competition[1–10]. Cell competition was originally found in *Drosophila*[11], but recent studies have revealed that cell competition can also occur in vertebrates[12,13]. For instance, it has been shown in mammalian cell culture systems that when Ras-, Src- or ErbB2-transformed cells are surrounded by normal cells, the transformed cells are apically extruded and leave the epithelium[14–17]. Apical extrusion of transformed cells has been also observed in zebrafish embryos and in various epithelial tissues in mice[15,18,19]. Importantly, when transformed cells alone are present, apical extrusion does not occur, implying that there are certain types of cell–cell communication between normal and transformed cells. During the process of apical extrusion, cytoskeletal proteins such as filamin accumulate in normal cells at the interface with transformed cells, thereby exerting physical forces that are required for the elimination of transformed cells[20]. Thus normal cells are able to sense the presence of the neighbouring transformed cells and actively eliminate them from epithelia, a process termed 'epithelial defence against cancer' (EDAC). Despite this anti-tumour activity, cancer often arises from epithelial tissues, suggesting that there may be molecular mechanisms that diminish EDAC. However, inhibitory molecular machinery for apical eradication of transformed cells remains largely unknown.

Cyclooxygenase (COX) is one of the key mediators in inflammation. It catalyses the conversion from arachidonic acid to prostaglandin (PG) $H_2$, which is further converted to various PGs or thromboxane. COX family is comprised of three members, and COX-1 and COX-2 play major physiological and pathological roles[21,22]. Constitutively expressed COX-1 is involved in tissue homoeostasis. In contrast, expression of COX-2 is stimulated by various proinflammatory cytokines, leading to synthesis of PGs. Among COX-2-mediated PGs, $PGE_2$ is a versatile lipid mediator involved in a variety of physiological and pathological processes, such as dilation of blood vessels and increased microvascular permeability[23–25].

In this study, we demonstrate that COX-2 expression and $PGE_2$ production are upregulated in normal cells neighbouring RasV12-transformed cells in a non-cell-autonomous fashion, which plays an inhibitory role in apical extrusion of transformed cells from epithelia.

## Results

### COX-2 expression is increased in mix-cultured normal cells.
To investigate whether and how the presence of RasV12-transformed cells influences the gene expression profile of the neighbouring normal epithelial cells, we performed microarray analysis. First, normal Madin–Darby canine kidney (MDCK) cells were co-cultured with MDCK cells expressing green fluorescent protein (GFP) or GFP-RasV12 (Fig. 1a). GFP-negative normal MDCK cells were then collected by fluorescence-activated cell sorting (FACS), and expression of various genes was compared between the two conditions (Fig. 1a). Expression of a number of genes was upregulated or downregulated in normal cells co-cultured with GFP-RasV12 cells, compared with those co-cultured with GFP cells (Fig. 1b). Among them, the expression of the *PTGS2* gene encoding COX-2 was most profoundly upregulated (Fig. 1b). The non-cell-autonomous upregulation of COX-2 was also confirmed by quantitative polymerase chain reaction (qPCR; Fig. 1c). Comparable upregulation of COX-2 expression was also observed in normal cells co-cultured with Src-transformed cells (Supplementary Fig. 1a). Furthermore, we showed by western

blotting and immunofluorescence that the protein level of COX-2 was also upregulated in normal cells co-cultured with RasV12-transformed cells (Fig. 1d–f and Supplementary Fig. 1b). Collectively, these data indicate that the presence of RasV12 cells augments the expression of COX-2 in the surrounding normal epithelial cells in a non-cell-autonomous fashion.

The expression of COX-2 can be regulated by a variety of stimuli or signalling pathways[26,27]. We then explored the upstream signalling pathways/molecules of the COX-2 expression by examining the effect of various inhibitors. Addition of inhibitors for nuclear factor-κB, activator protein-1, cAMP response element-binding protein or p38 mitogen-activated protein kinase did not significantly affect the expression level of COX-2 (Supplementary Fig. 2a–d). By contrast, addition of the protein kinase C (PKC) inhibitor bisindolylmaleimide (BIM)-I substantially suppressed the COX-2 expression (Fig. 2a). Furthermore, to detect PKC activity, we used an antibody against a phosphorylated consensus sequence recognized by PKC. Immunofluorescence analysis showed that activity of PKC was elevated in normal cells co-cultured with RasV12 cells, which was diminished by BIM-I (Fig. 2b). These results suggest that PKC acts upstream of COX-2.

### COX inhibitor or COX-2 knockout promotes apical extrusion.
Next, we examined a functional role of COX-2 in cell competition. In previous studies, we have demonstrated that RasV12-transformed MDCK cells are apically extruded when they are surrounded by normal MDCK cells[14,20]. Ibuprofen suppresses the activity of COX-1 and COX-2, whereas lumiracoxib suppresses COX-2 activity. We found that addition of either ibuprofen or lumiracoxib significantly elevated the apical extrusion ratio of RasV12 cells (Fig. 3a). We then established COX-2-knockout normal or RasV12-transformed MDCK cell lines (Supplementary Fig. 3a, b). When RasV12 cells were surrounded by COX-2-knockout normal cells, apical extrusion was profoundly enhanced (Fig. 3b). In contrast, knockout of COX-2 in RasV12 cells did not affect apical extrusion (Fig. 3c). Collectively, these data suggest that COX-2 in the surrounding normal cells negatively regulates apical elimination of RasV12-transformed cells.

### $PGE_2$ plays a negative role in apical extrusion.
To further understand the involvement of lipid metabolites in cell competition, we performed a comprehensive quantitative lipidomics analysis using conditioned media from three different culture conditions: normal cells alone, RasV12-transformed cells alone, and mix culture of normal and RasV12 cells. Among the detected lipid metabolites, the amount of $PGE_2$ was most significantly upregulated in the mix culture condition, compared with that in alone culture conditions (Fig. 4a and Table 1). COX-2 catalyses the conversion from arachidonic acid to $PGH_2$ (through $PGG_2$), and $PGH_2$ is further converted to various PGs and thromboxane, among which the role of $PGE_2$ in inflammation has been intensively studied[23–25]. Using the high-sensitivity enzyme-linked immunosorbent assay (ELISA) kit, we confirmed that the $PGE_2$ production was elevated in the mix culture condition (Fig. 4b). The level of $PGE_2$ in the mix culture condition was profoundly diminished by COX inhibitor or COX-2 knockout in normal cells (Fig. 4b and Supplementary Fig. 4a), suggesting that the production of $PGE_2$ is mainly mediated by COX-2 in normal cells under the mix culture condition. In addition, $PGE_2$ treatment attenuated apical extrusion of RasV12-transformed cells in a dose-dependent manner (Fig. 4c, d). MF63 selectively inhibits microsomal PGE synthase-1, which catalyses the conversion from $PGH_2$ to $PGE_2$. Addition of MF63 strongly suppressed the production of $PGE_2$ (Fig. 4e) and significantly promoted apical extrusion of RasV12 cells (Fig. 4f). Furthermore, $PGE_2$ treatment

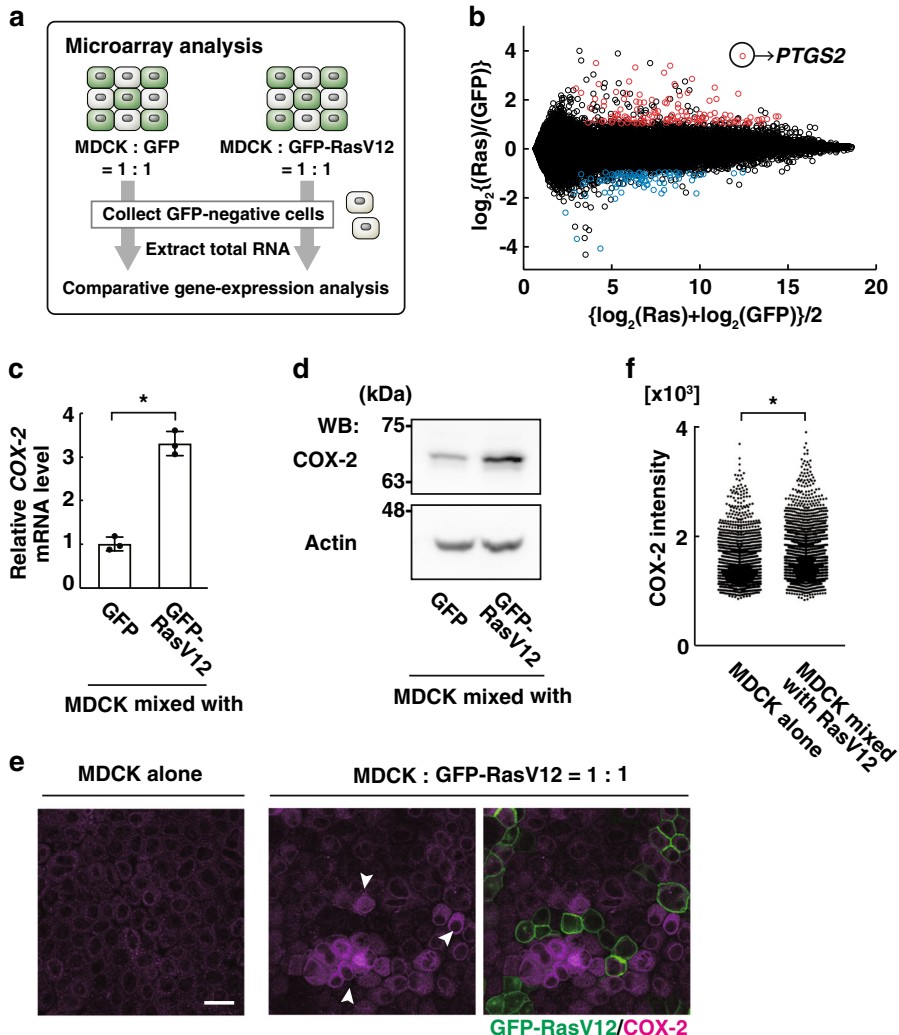

**Fig. 1 Expression of COX-2 is non-cell-autonomously elevated in normal epithelial cells co-cultured with RasV12-transformed cells. a** Schematics of microarray analysis. Normal MDCK cells were co-cultured with GFP- or GFP-RasV12-expressing MDCK cells. GFP-negative normal MDCK cells were then collected by FACS sorting, and the extracted total RNAs were subjected to comparative gene expression analysis between the two culture conditions. **b** Graphic display of expression profiling data of the microarray analysis. The vertical axis is the $\log_2$-ratio (mixed with Ras vs with GFP), while the horizontal axis represents the average log values. Red or blue dots indicate genes of which expression is significantly and more than twofold upregulated or downregulated, respectively, in the mix culture with Ras cells. The $P$ value for $PTGS2$ is $1.8 \times 10^{-4}$ (Student's $t$ test). **c, d** Quantitative RT-PCR (**c**) or western blotting analysis (**d**) of COX-2 expression. Cell lysates from FACS-sorted GFP-negative normal cells were examined. **c** Data are mean ± s.d. from three independent experiments. Values are expressed as a ratio relative to GFP. *$P = 0.0019$ (Student's $t$ test). **e, f** Immunofluorescence analysis of COX-2 expression. MDCK cells were cultured alone or co-cultured with MDCK GFP-RasV12 cells at a ratio of 1:1, followed by immunofluorescence analysis with anti-COX-2 antibody. **e** Arrowheads indicate COX-2-positive cells. Scale bars, 20 μm. (**f**) $n = 2152$ cells (alone) or 2321cells (with RasV12) from two independent experiments. *$P < 1.0 \times 10^{-4}$ (Mann–Whitney test).

substantially diminished the accumulation of filamin, a positive regulator for apical extrusion, in normal cells that surrounded RasV12 cells (Fig. 4g, h). Previous studies have demonstrated that high cell density conditions can induce apical extrusion within epithelia[28,29]. We found that $PGE_2$ treatment did not significantly affect the frequency of the crowded cell extrusion (Supplementary Fig. 4b–d). Collectively, these data indicate that $PGE_2$ production in normal cells negatively regulates apical elimination of RasV12-transformed cells by suppressing EDAC.

We further explored downstream of $PGE_2$. $PGE_2$ can bind and activate G-protein-coupled receptors EP (E prostanoid receptor) 1–4, thereby activating various downstream mediators[30]. Agonist for EP2 or EP4 suppressed apical extrusion of RasV12 cells, whereas EP1 or EP3 agonist had no effect (Supplementary Fig. 5a). However, antagonist for the respective receptor did not

significantly affect apical extrusion (Supplementary Fig. 5b). In addition, knockout of EP2 in normal or RasV12 cells did not influence the frequency of apical extrusion (Supplementary Fig. 5c–e). These results imply that multiple EP receptors or intracellular $PGE_2$ might regulate the downstream pathways (please see the 'Discussion' section for further discussion on upstream and downstream of the COX-2-$PGE_2$ pathway).

**Inflammation suppresses apical extrusion in vivo.** To examine the functional involvement of COX-2 in the eradication of transformed cells in vivo, we used a cell competition mouse model system. A CK19-Cre-ERT2 mouse was crossed with an LSL-eYFP or LSL-RasV12-IRES-eGFP mouse (Fig. 5a). We then administered a low dose of tamoxifen, which induced

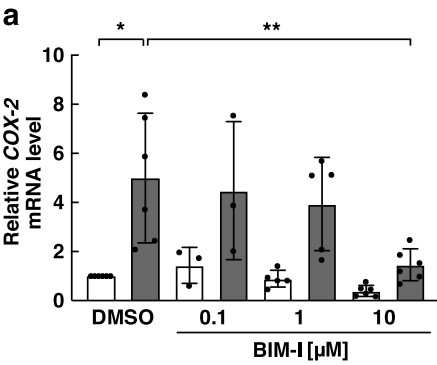

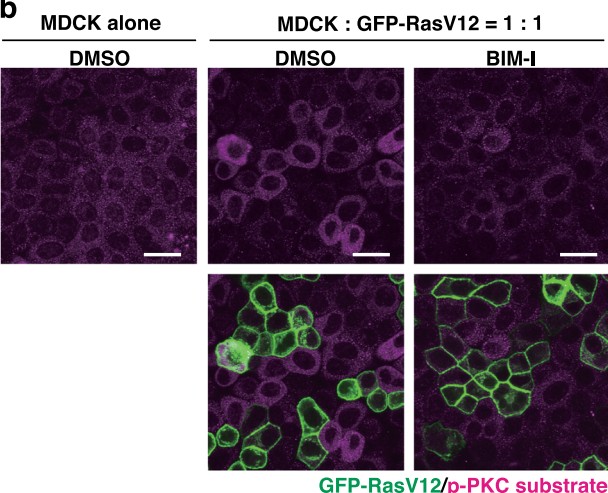

GFP-RasV12/p-PKC substrate

**Fig. 2 The PKC inhibitor BIM-I suppresses PKC activation and COX-2 expression in normal cells co-cultured with RasV12-transformed cells.** **a** Effect of BIM-I on the *COX-2* mRNA level in normal MDCK cells co-cultured with GFP-expressing MDCK cells (white) or GFP-RasV12-expressing MDCK cells (grey). Data are mean ± s.d. from six (DMSO, 10 μM BIM-I), five (1 μM BIM-I) or three (0.1 μM BIM-I) independent experiments. Values are expressed as a ratio relative to DMSO (MDCK mixed with GFP). *$P = 0.014$, **$P = 0.021$ (unpaired $t$ test). **b** Effect of BIM-I on PKC activation in normal cells co-cultured with RasV12 cells. MDCK cells were cultured alone or co-cultured with MDCK GFP-RasV12 cells at a ratio of 1:1 in the presence or absence of BIM-I (10 μM), followed by immunofluorescence analysis with anti-p-PKC substrate antibody. Scale bars, 20 μm.

recombination events less frequently, resulting in the expression of RasV12 in a mosaic manner within various epithelial tissues[19]. In this study, we focussed on the pancreas, as Ras mutations are frequently observed in human pancreatic cancer. By immuno-fluorescence analysis, we found that the expression of COX-2 was elevated in pancreatic epithelia harbouring RasV12-expressing cells (Fig. 5b). Accordingly, expression levels of a variety of inflammatory marker proteins were elevated in the RasV12-expressing pancreas compared with the YFP only-expressing pancreas, which was markedly reduced by ibuprofen treatment (Fig. 5c, d). RasV12-expressing cells were often apically eliminated from the pancreatic epithelial duct, but ibuprofen treatment further promoted apical extrusion of RasV12 cells (Fig. 5e–g). Collectively, these results suggest that increased expression of COX-2 accompanied by RasV12 induction suppresses apical elimination of transformed cells.

We further investigated the effect of extrinsically mediated inflammation on apical extrusion of RasV12-transformed cells. Caerulein is a cholecystokinin analogue that induces an infiltration of inflammatory cells into the pancreas leading to pancreatitis[31–34]. Caerulein was continuously administered for three weeks to induce chronic pancreatitis, and tamoxifen was then injected to induce RasV12 expression, followed by caerulein treatment for two more weeks (Fig. 6a). We confirmed that the experimental condition indeed induced inflammatory responses (Fig. 6b and Supplementary Fig. 6a). In control mice, the number of yellow fluorescent protein (YFP) only-expressing cells was not affected by caerulein (Supplementary Fig. 6b). In contrast, the number of remaining RasV12 cells within pancreatic tissues was substantially increased by caerulein treatment (Fig. 6c, d). This result indicates that extrinsically induced inflammation can also suppress elimination of RasV12-transformed cells from epithelial tissues.

## Discussion
Previous studies in *Drosophila* and mammals have revealed that during the event of cell competition, expression and activity of a variety of molecules/signalling pathways are modulated in both normal and transformed cells in a non-cell-autonomous fashion. Those changes often positively regulate the elimination of trans-formed cells. For instance, various positive regulators of cell competition have been found in *Drosophila*, including Flower, Spätzle/Toll signalling and Sas/PTP10D[35–37]. A role for Flower as positive regulator is conserved in mammals as well[38,39]. In mammals, Warburg-effect-like metabolic changes occur in trans-formed cells surrounded by normal cells, thereby promoting apical extrusion of transformed cells[18]. In normal cells, cytoskeletal proteins filamin and vimentin accumulate at the interface with transformed cells, which generate physical forces that are required for apical extrusion[20]. On the other hand, it remains elusive whether and how cell competition is also governed by negative regulators. In *Drosophila*, the expression of a soluble factor Sparc is upregulated in loser cells, which inhibits caspase activation of loser cells in a self-protective manner[40]. In this study, we demonstrate that the COX-2-PGE₂ pathway acts as a negative regulator of cell competition in winner cells. The expression level of COX-2 can be influenced by a number of upstream signalling pathways; thus the inhibitory role of COX-2 would be upregulated or downregulated depending on external stimuli or environments. Hence, under various conditions, the consequence of cell com-petition might be controlled by the power balance between posi-tive and negative regulators.

Expression of COX-2 is transcriptionally upregulated in nor-mal epithelial cells that surround RasV12-transformed cells. Among the tested inhibitors, the pan-PKC inhibitor BIM-I suppresses the non-cell-autonomous upregulation of COX-2, suggesting that PKC plays a positive role in COX-2 expression. However, previous studies have demonstrated that PKC inhibitor moderately suppresses apical extrusion of transformed cells[20,41]. Moreover, PKC-ε is accumulated in normal cells surrounding transformed cells, which induces accumulation of vimentin filaments, a positive regulator for apical extrusion[20]. Therefore, it is plausible that PKC phosphorylates multiple positive or nega-tive regulators of cell competition, probably at different stages of apical extrusion; temporal and dynamic regulation of PKC activity can thus modulate elimination of transformed cells positively or negatively. In future studies, PKC-catalysed phos-phorylation of those cell competition regulators needs to be further examined.

COX-2 promotes the production of various prostaglandins, including PGE₂. We present several lines of evidence indicating

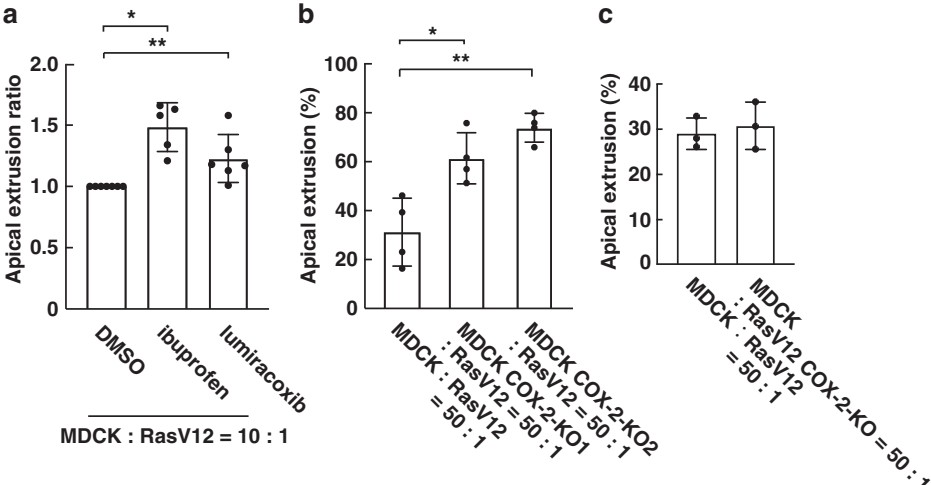

**Fig. 3 COX inhibitor or COX-2 knockout in normal cells promotes apical extrusion of RasV12-transformed cells. a** Effect of the COX inhibitor ibuprofen (10 μM) or lumiracoxib (10 μM) on apical extrusion of RasV12 cells. Data are mean ± s.d. from seven (DMSO), five (ibuprofen) or six (lumiracoxib) independent experiments. Values are expressed as a ratio relative to DMSO. *$P$ = 0.0054, **$P$ = 0.0036 (unpaired $t$ test). **b**, **c** Effect of COX-2-knockout in normal cells (**b**) or RasV12 cells (**c**) on apical extrusion of RasV12 cells. **b** Data are mean ± s.d. from four independent experiments. *$P$ = 0.015, **$P$ = 0.0041 (Student's $t$ test). **c** Data are mean ± s.d. from three independent experiments.

that the COX-2-PGE$_2$ pathway plays a crucial role in cell competition between normal and transformed cells. First, the amount of PGE$_2$ is elevated in the mix culture of normal and RasV12-transformed cells. Second, COX-2 knockout in normal cells diminishes the production of PGE$_2$. Third, addition of PGE$_2$ profoundly suppresses apical extrusion of RasV12 cells. Fourth, PGE$_2$ synthase inhibitor treatment promotes apical extrusion of RasV12 cells. PGE$_2$ can bind and activate EP 1–4. However, addition of antagonists for the respective receptors does not significantly affect apical extrusion. In addition, EP2 knockout in normal or RasV12 cells does not influence the frequency of apical extrusion. It is possible that multiple PGE$_2$ receptors on normal or RasV12 cells regulate the process of apical extrusion in a concerted manner. Alternatively, not only extracellular PGE$_2$ but also intracellular PGE$_2$ might modulate signalling pathways that affect apical extrusion[42–44]. It has been reported in *Drosophila* that various proinflammatory pathways are activated in loser cells, leading to their elimination from tissues[36,45–49]. Our study thus provides another dimension for the role of inflammation in cell competition.

At the interface between normal and transformed epithelial cells, the COX-2-PGE$_2$ pathway is upregulated in normal epithelial cells in a non-cell-autonomous fashion, which suppresses apical extrusion of transformed cells. In addition to this intrinsic inflammatory response, our data suggest that extrinsically mediated inflammation can also suppress cell competition; either addition of PGE$_2$ or caerulein treatment diminishes the frequency of apical elimination of transformed cells from epithelia (Supplementary Fig. 6c). We also demonstrate that administration of non-steroidal anti-inflammatory drug (NSAID) promotes apical extrusion of RasV12-transformed cells from epithelial layers both in vitro and in vivo. Inflammation is considered to be one of the crucial tumour-promoting factors[50–52]. Chronic inflammatory diseases increase the risk of certain types of cancer, such as colon and pancreas. Epidemiological evidence indicates that NSAID treatment significantly decreases the incidence of various types of cancer, including colon and pancreatic cancer[53,54]. It is generally believed that inflammation influences tumour progression at the mid-stage or late stage of carcinogenesis. However, the data in this

study imply that inflammation can also facilitate tumour initiation by diminishing EDAC at the initial stage of carcinogenesis.

## Methods

**Antibodies, materials and plasmids.** The following primary antibodies were used in this study: mouse anti-β-actin (MAB1501R clone C4) and mouse anti-GAPDH (MAB374) antibodies from Merck Millipore, mouse anti-filamin (F6682) and mouse anti-vinculin (V44505) antibodies from Sigma-Aldrich, rabbit anti-COX-2 (ab15191) and chicken anti-GFP (ab13970) antibodies from Abcam, rabbit anti-Phospho-(Ser) PKC Substrate (#2261) antibody from Cell Signaling Technology, rat anti-E-cadherin (131900) antibody from Life Technologies, and allophycocyanin-conjugated CD45.2 (20–0454) antibody from TONBO. Peroxidase-conjugated AffiniPure anti-mouse and anti-rabbit IgG were from Jackson ImmunoResearch. Alexa-Fluor-568- and -647-conjugated phalloidin (Life Technologies) were used at 1.0 U ml$^{-1}$. Alexa-Fluor-568- and −647-conjugated secondary antibodies were from Life Technologies. Alexa-Fluor-488-conjugated secondary antibody was from Abcam. Hoechst 33342 (Life Technologies) was used at a dilution of 1:5000 for immunofluorescence. Type-I collagen (Cellmatrix Type I-A) was obtained from Nitta Gelatin and was neutralized on ice to a final concentration of 2 mg ml$^{-1}$ according to the manufacturer's instructions. To fluorescently stain living MDCK-pTR GFP or MDCK-pTR GFP-RasV12 cells, CMFDA (green dye) (Life Technologies) was used according to the manufacturer's instructions. The following regents were obtained from Cayman Chemical: PGE$_2$, agonists selective for each EP subtype (Iloprost, Butaprost (free acid), Sulprostone and L-902,688 for EP1, EP2, EP3 and EP4, respectively) and antagonists selective for each EP receptor (ONO-8711 and PF-04418948 for EP1 and EP2, respectively). L-798106 (EP3 antagonist) was obtained from Sigma-Aldrich. ONO-AE3–208 (EP4 antagonist) was obtained from ChemScene. The following COX inhibitors, ibuprofen, ibuprofen sodium salt and lumiracoxib were from Sigma-Aldrich. Ibuprofen and ibuprofen sodium salt were used for cell culture and mice, respectively. MF63 was from ChemScene. BIM-I was from Calbiochem. SB203580 (10 μM) and SR11302 (5 μM) were from TOCRIS. BAY117082 (1 μM) and C646 (1 μM) were from Santa Cruz Biotechnology. The SiR-actin Kit (far-red silicon rhodamine (SiR)-actin fluorescence probe) was obtained from SPIROCHROME for staining F-actin without washing procedure and was used according to the manufacturer's instructions.

**Cell culture.** MDCK cell lines were used in this study. The parental MDCK cells were a gift from W. Birchmeier[55]. Mycoplasma contamination was regularly tested for all cell lines in use using a commercially available kit (MycoAlert, Lonza). MDCK cells stably expressing GFP (MDCK-pTR GFP), GFP-RasV12 (MDCK-pTR GFP-RasV12) or GFP-cSrcY527F (MDCK-pTR GFP-cSrcY527F) in a tetracycline-inducible manner were established and cultured as previously described[14,56]. To establish CRISPR/Cas9-mediated COX-2- or EP2-knockout cells, guide sequences of COX-2 or EP2 single-guide RNA (sgRNA) targeting *Canis* COX-2 or EP2 were designed on exons 1 as described previously[57]. COX-2 sgRNA sequence (COX-

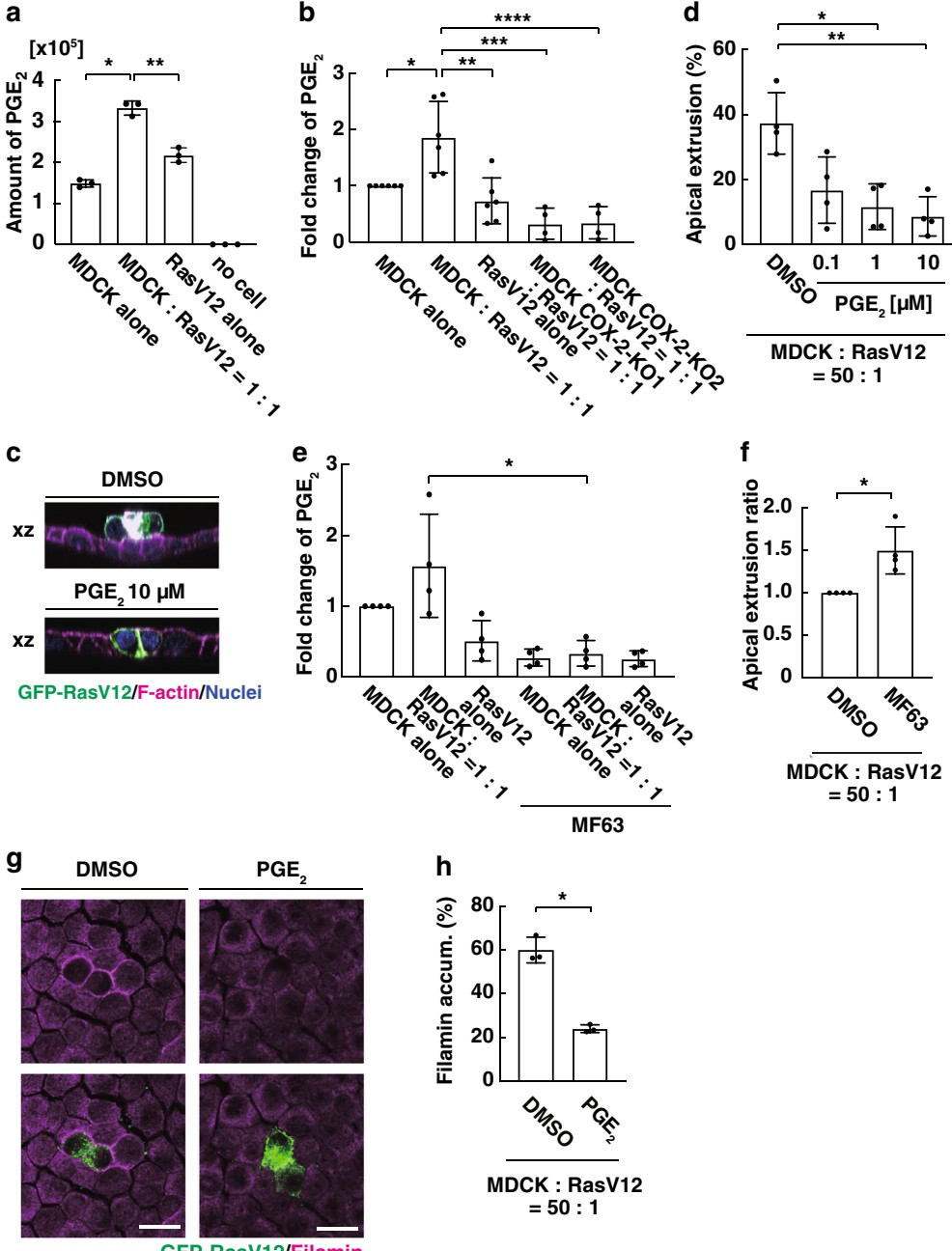

**Fig. 4 PGE$_2$ negatively regulates apical extrusion of RasV12-transformed cells that are surrounded by normal cells. a, b** Measurement of PGE$_2$ in conditioned media under the indicated culture conditions by comprehensive quantitative lipidomics analysis (**a**) or using the high-sensitivity ELISA kit (**b**). **a** The values in the vertical axis indicate the chromatographic peak area of the mass spectrometric analysis. Data are mean ± s.d. from triplicate samples. *$P = 0.0095$, **$P = 0.0013$ (Student's $t$ test). **b** Data are mean ± s.d. from six (left three) and four (right two) independent experiments. Values are expressed as a ratio relative to MDCK alone. *$P = 0.020$, **$P = 0.0056$ (Student's $t$ test), ***$P = 0.0011$, ****$P = 0.017$ (unpaired $t$ test). **c, d** Effect of PGE$_2$ on apical extrusion of RasV12 cells. **c** The xz-immunofluorescence images of GFP-RasV12-expressing cells surrounded by normal cells in the presence or absence of PGE$_2$. Scale bar, 20 μm. **d** Data are mean ± s.d. from four independent experiments. *$P = 0.044$, **$P = 0.018$ (Student's $t$ test). **e** Effect of the PGE synthase-I inhibitor MF63 (0.1 μM) on the production of PGE$_2$. Data are mean ± s.d. from three independent experiments. Values are expressed as a ratio relative to MDCK alone. *$P = 0.021$ (Student's $t$ test). **f** Effect of MF63 (0.1 μM) on apical extrusion of RasV12 cells. Data are mean ± s.d. from three independent experiments. Values are expressed as a ratio relative to DMSO. *$P = 0.035$ (Student's $t$ test). **g, h** Effect of PGE$_2$ (10 μM) on filamin accumulation in normal cells at the interface with the neighbouring RasV12 cells. **g** Immunofluorescence images of filamin in the presence or absence of PGE$_2$. Scale bars, 20 μm. **h** Data are mean ± s.d. from three independent experiments. *$P = 0.011$ (Student's $t$ test).

**Table 1 Comprehensive quantitative lipidomics analysis using liquid chromatography–selected reaction monitoring–mass spectrometry (LC-SRM-MS).**

| | MDCK alone | RasV12 alone | Average of alone cultures | Mix culture | P value |
|---|---|---|---|---|---|
| Prostaglandin $E_2$ | 149,032.0 | 218,274.7 | 183,653.3 | 333,123.3 | 0.00011 |
| Prostaglandin $A_2$ | 6590.7 | 8940.3 | 7765.5 | 13,786.3 | 0.00034 |
| 17-Hydroxy docosahexaenoic acid | 1098.0 | 2104.7 | 1601.3 | 3390.7 | 0.013 |
| 11-HETE | 1476.7 | 5309.0 | 3392.8 | 6840.3 | 0.013 |
| 8-Iso prostaglandin $E_2$ | 5696.7 | 6452.7 | 6074.7 | 11,690.7 | 0.043 |
| 6-Keto prostaglandin $F_{1\alpha}$ | 5651.0 | 11,411.3 | 8531.2 | 12,791.0 | 0.048 |
| Prostaglandin $F_{2\alpha}$ | 7913.0 | 22,378.7 | 15,145.8 | 22,196.0 | 0.087 |
| Oleoyl ethanolamide | 1979.3 | 1353.7 | 1666.5 | 4346.7 | 0.132 |
| Prostaglandin $D_2$ | 1260.3 | 6405.3 | 3832.8 | 4835.0 | 0.474 |

The values for MDCK alone, RasV12 alone or Mix culture indicate the mean of chromatographic peak area for the corresponding SRM transitions from the triplicate samples. P values between 'Average of alone cultures' and 'Mix culture' were calculated using unpaired t test.

2 sgRNA, 5'-CGCCCTGGTGCTCTGCGCC-3') or EP2 sgRNA sequence (EP2 sgRNA1, 5'-CATCGCGCTGGCGCTGCTG-3' or EP2 sgRNA2, 5'-GCTGC TGGCGCGCCGCTGG-3') was introduced into the pCDH-QC-sgRNA control (sgControl) vector[58] using primers listed in Supplementary Table 1. First, MDCK cells were infected with lentivirus carrying pCW-Cas9 as described[58] and were cultured in the 500 ng ml$^{-1}$ puromycin-containing medium. The tetracycline-inducible MDCK-Cas9 cells were transfected with pCDH-EF1-COX-2 sgRNA, pCDH-EF1-EP2 sgRNA1 or 2 by nucleofection, followed by selection in the medium containing 200 µg ml$^{-1}$ of hygromycin, and subjected to limiting dilution. Indels on the COX-2 or EP2 exons in each monoclone were analysed by direct sequencing using primers listed in Supplementary Table 1. To generate MDCK-pTRE3G GFP-RasV12-COX-2-KO or -EP2-KO cells, complementary DNA of GFP-H-RasV12 was cloned into BamHI/EcoRI sites of pPB-TRE3G-MCS-CEH-rtTA3-IP[18]. pPB-TRE3G GFP-H-RasV12 was introduced into the COX-2- or EP2-deleted cells by nucleofection and antibiotic selection (blasticidin, 5 µg ml$^{-1}$).

**Microarray analysis**. In all, $9.0 \times 10^6$ of 1:1 mix culture of MDCK and MDCK-pTR GFP cells or 1:1 mix culture of MDCK and MDCK-pTR GFP-RasV12 cells were cultured in non-coat 10-cm plastic dishes. After incubation with tetracycline for 16 h, following trypsin treatment, GFP-negative cells were collected by an analytical flow cytometer FACS Aria$^{TM}$ II. Total RNA was extracted from the isolated cells using TRIzol (Thermo Fisher Scientific) and an RNeasy Mini Kit (QIAGEN). The gene expression profiling was performed using the Canine (V2) Gene Expression Microarray Kit (Agilent Technologies) by Oncomics Co., Ltd. For analysis of the microarray data, the canine gene database canFam2:V2.0:May2005 was used.

**Quantitative real-time PCR**. For cultured cells, the purified total RNA was obtained from GFP-negative cells as described above. For RNA isolation from murine pancreas, pancreas was cut into small pieces using scissors and incubated in 2 mg ml$^{-1}$ collagenase (Wako) in the RPMI1640 medium (Sigma-Aldrich) containing 2% (vol/vol) new-born calf serum (Equitech-Bio) for 30 min at 37 °C with stirring. The cell suspensions were filtered through cell strainers (pore size, 70 µm; BD Biosciences) and used for gene expression analysis. The total RNA was extracted from the isolated cells using TRIzol and an RNeasy Mini Kit (QIAGEN) and reverse-transcribed using a QuantiTect Reverse Transcription Kit (QIAGEN). GeneAce SYBR qPCR Mix (NIPPON GENE) was used to perform qPCR using the StepOne system (Thermo Fisher Scientific). The primer sequences used are listed in Supplementary Table 1. We used *GAPDH* or *RPL13A* as a reference gene to normalize data.

**Immunofluorescence**. For immunofluorescence of cultured cells, MDCK-pTR GFP-RasV12 cells were mixed with MDCK cells at a ratio of 1:50, 1:10 or 1:1 and cultured on the collagen matrix as previously described[14]. The mixture of cells was incubated for 8–12 h until they formed a monolayer, followed by tetracycline treatment for 6 h (for immunofluorescence analyses of p-PKC substrate), 16 h (for immunofluorescence analyses of COX-2 or filamin) or 24 h (for apical extrusion analyses). Cells were fixed with 4% paraformaldehyde (PFA) (Sigma-Aldrich) in phosphate-buffered saline (PBS) and permeabilized with 0.5% Triton X-100 in PBS, except for filamin immunofluorescence where cells were fixed in methanol at −20 °C for 2.5 min, followed by blocking with 1% bovine serum albumin in PBS. Primary or secondary antibodies were incubated for 2 or 1 h, respectively, at ambient temperature. Alexa-Fluor-568- or -647-conjugated phalloidin was incubated for 1 h at ambient temperature. For immunohistochemical examinations of the pancreas, the mice were perfused with 1% PFA, and the isolated tissues were

fixed with 1% PFA in PBS for 24 h and embedded in FSC 22 Clear Frozen Section Compound (Leica Biosystems). Then, 10-mm-thick frozen sections were cut on a cryostat. The sections were blocked with Block-Ace (DS Pharma Biomedical) and 0.1% Triton X-100 in PBS. Primary or secondary antibodies were incubated for 2 or 1 h, respectively, at ambient temperature. Immunofluorescence images of cultured cells and mouse tissues were acquired using the Olympus FV1000 system and Olympus FV10-ASW software. For immunostaining analyses, we captured images using the same set of parameters (e.g. Scan Speed/Averaging, Laser Power, Sampling Frequency, Pinhole, Detector Setting) under each experimental setting to confirm the reproducibility of the obtained results. All primary antibodies were used at 1:100, except for mouse tissues: anti-GFP (1:1000), anti-E-cadherin (1:2000) and anti-CD45.2 (1:1000) antibodies. All secondary antibodies were used at 1:200 for cultured cells and at 1:1000 for mouse tissues.

**Western blotting**. Western blotting was performed as previously described[59]. Primary antibodies were used at 1:1000 except anti-GAPDH at 1:2000. Western blotting data were analysed using ImageQuant$^{TM}$ LAS4010 (GE Healthcare).

**Lipidomics analysis**. In all, $1.0 \times 10^6$ MDCK cells, MDCK-pTR GFP-RasV12 cells or 1:1 mix culture of MDCK and MDCK-pTR GFP-RasV12 cells were plated into non-coat plastic dishes (6-well, greiner) (triplicate under each condition). After incubation for 12 h, tetracycline was added to induce GFP-RasV12 expression, followed by incubation for 16 h. The culture media were then changed to foetal calf serum (FCS)-free Dulbecco's modified Eagle's medium with tetracycline, which were collected after 6 h and stored at −80 °C. After thawing, the samples were centrifuged for 10 min at $10,000 \times g$ and were purified with solid phase extraction with an Oasis HLB cartridges (Milford, MA, USA) as previously described[60]. Liquid chromatography–selected reaction monitoring–mass spectrometry (LC-SRM-MS) analysis was performed using a Nexera UHPLC system and a triple quadrupole mass spectrometer LCMS-8040 (Shimadzu, Kyoto, Japan) with a reversed-phase column (Kinetex C8, $2.1 \times 150$ mm, 2.6 µm; Phenomenex, Torrance, CA) as described previously[61]. LabSolutions software (Shimadzu) was used for peak detection and integration. Chromatographic peak area for the corresponding SRM transitions were used for comparison of lipid mediators.

**Measurement of PGE$_2$ concentration**. For measurement of PGE$_2$ concentration, $3.25 \times 10^5$ MDCK cells, MDCK-pTR GFP-RasV12 cells or 1:1 mix culture of MDCK and MDCK-pTR GFP-RasV12 cells were cultured into non-coat plastic dishes (24-well, greiner). After 16 h tetracycline incubation with or without the indicated COX inhibitor, culture media were changed to FCS-free DMEM with tetracycline and incubated for 2 h. Culture media were then collected, and PGE$_2$ concentration was measured using the PGE$_2$ high-sensitivity ELISA kit (ENZO Life Sciences) according to the manufacturer's instructions.

**Mice**. All animal experiments were conducted under the guidelines by the Animal Care Committee of Hokkaido University. The animal protocols were reviewed and approved by the Hokkaido University Animal Care Committee (approval number 12–0116). Cytokeratin19 (CK19)-CreERT2 mice[62] were crossed with CAG-loxP-STOP-loxP-EYFP mice[63] or DNMT1-CAG-loxP-STOP-loxP-HRasV12-IRES-eGFP mice[18] to create CK19-YFP mice or CK19-RasV12-GFP mice, respectively. The CAG-loxP-STOP-loxP-EYFP mice were obtained from the Jackson Laboratory (JAX stock #006148). For analyses, we used 8–36-week-old C57BL/6 mice of either sex. For PCR genotyping of mice, primers listed in Supplementary Table 1 were used. Mice were given a single intraperitoneal injection of 2 mg of tamoxifen in

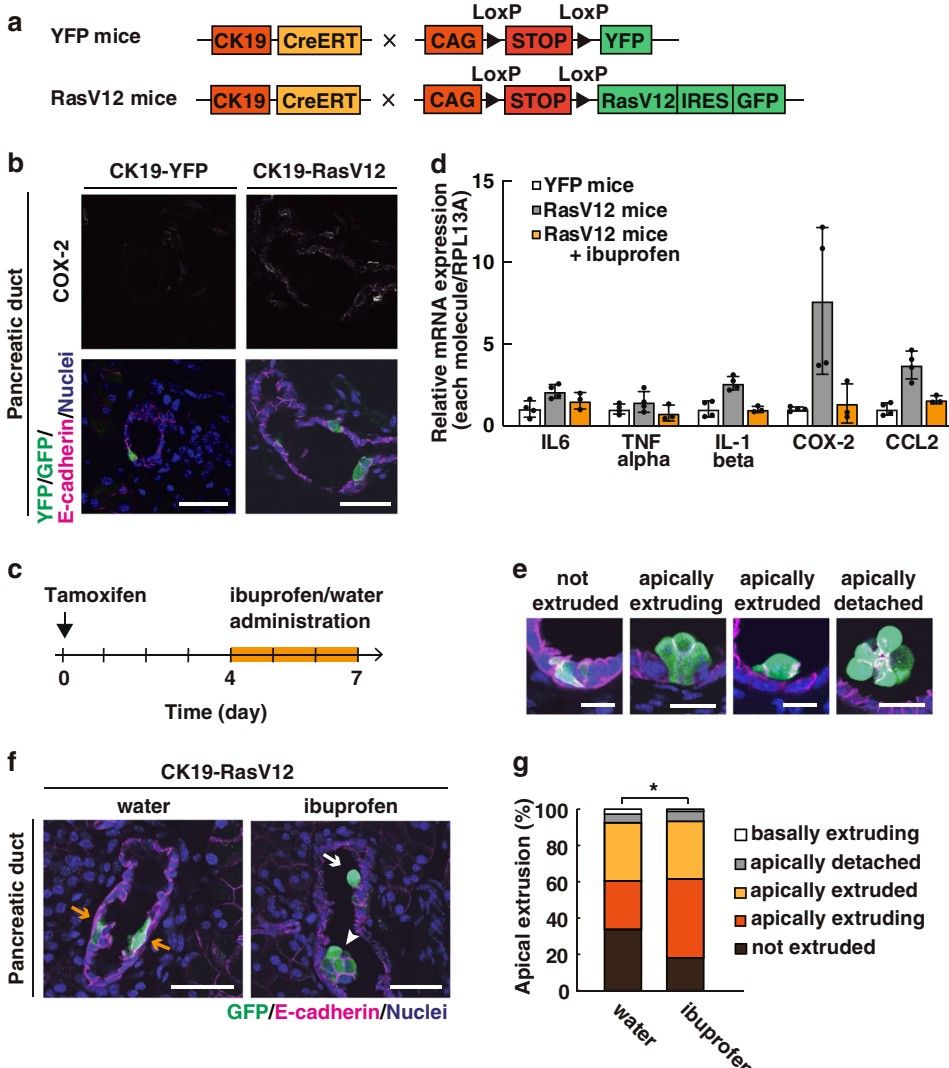

**Fig. 5 COX inhibitor treatment promotes apical elimination of RasV12-transformed cells from pancreatic ductal epithelia. a** Strategy for the establishment of the cell competition mouse model. **b** Immunofluorescence analysis for COX-2 in the pancreatic ducts harbouring YFP- or RasV12/GFP-expressing cells at 3 days after tamoxifen administration. Scale bars, 40 μm. **c** Experimental design for ibuprofen treatment. **d** mRNA expression of inflammatory factors in the RasV12-expressing pancreas with or without ibuprofen treatment. Tissue fractions of the pancreas were examined by quantitative RT-PCR. The mRNA expression level of the indicated inflammatory factors was normalized to that of *RPL13A*. Data are mean ± s.d. from five (YFP mice or RasV12 mice) or three (RasV12 mice+ibuprofen) independent experiments. **e–g** Effect of ibuprofen treatment on apical extrusion of RasV12-expressing cells from pancreatic epithelia. **e** Representative images of RasV12-expressing cells. 'not extruded': remaining within the epithelium. 'apically extruding': with their nucleus apically shifted, but still attached to the basement membrane. 'apically extruded': completely detached from the basement membrane and translocated into the apical lumen. 'apically detached': completely detached from the apical side of the epithelial layer. Scale bars, 20 μm. **f** Immunofluorescence images of pancreatic ducts at 7 days after tamoxifen administration with or without ibuprofen treatment. Yellow arrows: not extruded; white arrow: apically extruding; white arrowhead: apically extruded. Scale bars, 40 μm. **g** Quantification of apical extrusion of RasV12 cells from the pancreatic duct. $n = 174$ (water) or 234 (ibuprofen) from three mice. *$P = 0.020$ (chi-square test).

corn oil (Sigma-Aldrich) per 20 g of body weight for the induction of YFP or RasV12/GFP expression and then sacrificed at Day 3, Day 7 or Day 14 after Cre activation. To examine the effect of ibuprofen, ibuprofen sodium salt (Sigma) was dissolved at 1 mg ml⁻¹ in drinking water and then administered for 3 days before sacrifice. Acute pancreatitis was elicited by hourly (6 times) intraperitoneal injections of 50 μg kg⁻¹ body weight caerulein (Sigma-Aldrich, Tokyo, Japan), whereas control animals received a comparable amount of PBS. Mice were submitted to three episodes of caerulein treatment per week for 3 weeks[64], resulting in chronic pancreatitis.

**Statistics and reproducibility**. For data analyses, Mann–Whitney test, Chi-square test, unpaired *t* test or two-tailed Student's *t* test was used to determine *P* values as indicated in the figure legends. *P* values <0.05 were considered to be statistically significant. For quantification of apical extrusion of RasV12-transformed cells,

2–12 RasV12-transformed cells that were surrounded by the indicated cells were analysed. Filamin accumulation in the surrounding cells at the interface with RasV12-transformed cells was quantified for transformed cells of which >75% of the perimeter was surrounded by normal cells. When filamin accumulation was observed at more than one-third of the perimeter of the transformed cell, it was counted as filamin accumulation positive. More than 100 cells were analysed for apical extrusion, and >30 cells were analysed for filamin accumulation in each experiment. All immunofluorescence images were analysed by confocal microscopy. Immunofluorescence intensity of COX-2 was quantitatively analysed with a Cell Insight image cytometer (Thermo Fisher Scientific).

**Reporting summary**. Further information on research design is available in the Nature Research Reporting Summary linked to this article.

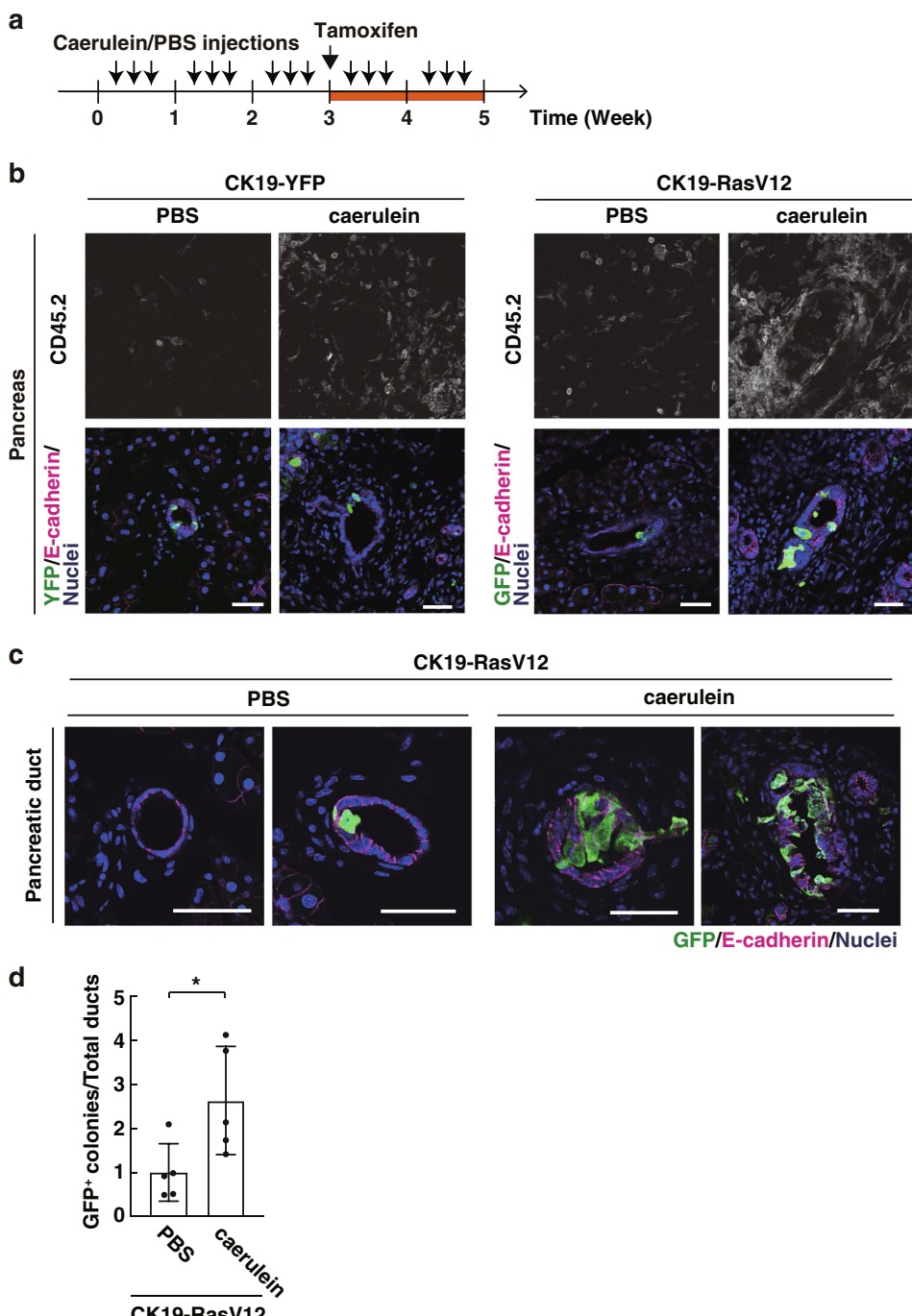

**Fig. 6 Caerulein-induced inflammation retards apical elimination of RasV12-expressing cells from pancreatic ductal epithelia. a** Experimental design for caerulein treatment. **b** Immunofluorescence analysis for the immune cell marker CD45.2 in the pancreatic tissues upon caerulein treatment. **c** Immunofluorescence images of RasV12/GFP-expressing cells with or without caerulein treatment. **b**, **c** Scale bars, 40 µm. **d** Quantification of RasV12-expressing cell groups (colonies) in the pancreatic epithelial ducts. The total number of analysed ducts are 963 (PBS) and 867 (caerulein). Values are expressed as a ratio relative to PBS. Data are mean ± s.d. from five mice. *$P = 0.039$ (Student's $t$ test).

## Data availability

Microarray data have been deposited in the ArrayExpress database at EMBL-EBI under accession number E-MTAB-8769. Source data are available in Supplementary Data 1. All other data sets generated during and/or analysed during the current study are available from the corresponding author on reasonable request.

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

## Acknowledgements

We thank Dr. Masaki Yamada for the assistance of lipidomics analyses. This work was supported by Japan Society for the Promotion of Science (JSPS) Grant-in-Aid for Scientific Research on Innovative Areas 26114001, the Scientific Research (A) 18H03994, Strategic Japanese-Swiss Science and Technology Program, AMED under Grant Numbers JP19ck0106361h0003 and JP19cm0106234h0002, SAN-ESU GIKEN CO. LTD and

the Takeda Science Foundation (to Y.F.) and Grant-in-Aid for JSPS Research Fellow JP19J10318 (to N.S.).

## Author contributions

N.S. designed experiments and generated most of the data. Y.Y. performed microarray screening and some of cell culture experiments. T.M. assisted in generation of COX-2-knockout cells. S.I. and K.K. assisted in experiments. S.M.T. and Y.K. performed lipidomics analyses. Y.F. conceived and designed the study. The manuscript was written by N.S. and Y.F. with assistance from the other authors.

## Competing interests

The authors declare no competing interests.
