## [Peer Review File · Communications Biology]

Reviewers' comments:

Reviewer #1 (Remarks to the Author):

In this manuscript, Nanami Sato and colleagues describe a new factor that can modulate the rate of extrusion of active Ras cells from epithelial layer both in vitro and in vivo. They show that COX2 is upregulated when Ras cells are mixed with WT cells, which leads to PGE2 pathway activation and a reduction of the rate of Ras cell elimination. While mechanistically this study does not necessarily provides much for the understanding of the recognition process of the Ras cells, this is very interesting physiologically speaking as it provides a very interesting link between inflammation and the probability of elimination of potential tumorigenic cells. Those results are also interesting as they illustrate how the same signals can activate both pro and anti competition signals, suggesting that the decision making process is complex and well regulated (as illustrated by many other studies in competition, as pointed by the authors in the discussion).

The manuscript is well written and the data very convincing. The validation of the mechanism in vivo clearly adds a lot of value to the manuscript. I have at this stage only minor points to raise that will help to clarify some points and strengthen some aspects of the demonstration.

1. While the manuscript provide some immunostaining of COX2 and other elements in the pathway, it was not clear to me to which extent does this activation happens. Could the authors provides larger view of the epithelial layer ? Is COX2 upregulation restricted to the neighbourhood of Ras cells or does it extend quite far away ?

2. A previous paper (including some authors of this study, Takanori Chiba et al., Scientific report, 2016) showed that PKC inhibition with BIM1 drugs suppresses Ras cell extrusion (Fig. 3a of this previous paper). This is in contradiction with the results of this new manuscript which suggests that PKC is required for COX2 expression, which inhibits Ras cells extrusion. This may not be so surprising as PKC was shown by the same lab to be required to activate Vimentin, a key component of the Ras cell extrusion (Kajita et al., Nat. Comm., 2014). It would argue again that the same pathway can both activate pro and anti competition components. I would strongly encourage the authors to include and explain this apparent contradiction in the discussion

3. At this stage, it is not clear whether this pathway is specific of the oncogenic cell extrusion. I don't know what is the basal rate of extrusion in homogenous MDCK layer, but it would be interesting to check if PGE2 treatment could also affect normal extrusion.

4. The results of this study suggest that inflammation would inhibit cell competition. Most of the data coming from Drosophila rather suggest that pro-inflammatory pathways are required for cell elimination during competition (e.g.: JAK-STAT pathway, work from Erika Bach lab and Eugenia Piddini lab, TNF/JNK, work from Igaki, Moreno and Piddini lab, Toll pathway, work from Johnston lab). I believe this may deserve a little paragraph in the discussion.

5. How comparable are the intensity of the immunostainings between different cultures (e.g.: Fig. 1e, 2b) ? Are the intensity levels reproducible for a given condition (provided that the acquisition parameters and contrast and maintained) ? This may deserve some explanation in the methods.

Reviewer #2 (Remarks to the Author):

The COX-2/PGE2 pathway suppresses apical elimination of RasV12-transformed cells from epithelia

The work described in this manuscript contributes to our understanding of Epithelial defense against cancer (EDAC), and under which circumstances EDAC fails to prevent tumorigenesis. The molecular

signaling that contributes to the breakdown of EDAC is not known, and remains a critical question in tumorigenesis. This study provides the first evidence that the COX-2-PGE2 pathway acts as a negative regulator of cell competition in winner cells and that inflammation, which is synonymous with cancer, suppresses EDAC. Importantly, the authors were also able to show that NSAIDs such as ibuprofen promote the apical extrusion of RasV12-transformed cells in both cell culture and mouse model of cell competition.

The findings presented in this manuscript address a major question in the field of cell competition and also suggest that inflammation determines cancer growth also in early stages of development. The authors identified COX2 signaling as major signaling mediator, however, the data presented herein do not fully describe PGE2-EP receptor engagement.

Fig 1: Identification of altered genes in normal cells following co-culture with Ras transformed cells
Authors performed microarray analysis of altered genes in normal cells following (MDCK) cells co-cultured with MDCK cells expressing GFP or GFP-RasV12 and found upregulation of COX2 gene (PTGS2) in normal cells and confirmed upregulation of this gene by qPCR.

Fig 2

Pan-PKC inhibitor BIM-I suppresses the non-cell-autonomous upregulation of COX-2 in normal cells. Results illustrated in this figure suggest that PKC acts upstream of COX-2, however, actual upstream regulators of COX2 are still unclear and some information there will be important to know.

Fig 3

Cox-2 negatively regulated apical elimination of RasV12 transformed cells by normal cells. Suppression of COX1/COX2 activity increases apical extrusion of RasV12 transformed cells as does surrounding Rasv12 transformed cells with COX2-KO normal cells.

Fig 4

Authors used comprehensive quantitative-lipidomics analysis using conditioned media and found significant upregulation of PGE2 in co-culture medium and confirmed PGE2 upregulation using ELISA. The authors next showed that PGE2 secretion negatively regulates apical extrusion of Ras V12 transformed cells in series of reproducible experiments. When PGE2 production was blocked, apical extrusion of Ras V12 cells was rescued.

However, it remains unknown how PGE2 engagement with which EP receptors or how intracellular PGE2 signaling mediates these effects.

Fig 5: Mouse model of cell competition

Focused on pancreatic epithelia, since Ras is common mutation in pancreatic cancer. COX-2 was elevated in pancreatic epithelia harbouring RasV12-expressing cells as well as other inflammatory markers. Results from in vivo experiments confirmed findings from cell culture models that COX2 negatively regulated apical elimination of transformed cells, but blocking COX2 signaling with NSAIDs enhances elimination of transformed cells.

Fig 6

Authors further studied role of inflammation in reducing efficacy of EDAC. Their major finding here was that treatment of mice with caerulein to induce pancreatitis followed by induction of Rasv12 transformed cells increased number of RasV12 transformed cells in pancreatic tissue.

Used 3 mice for these experiments, we feel this experiment can have n=5.

The authors used CD45.2 to prove inflammation, but we suggest other inflammation markers must be included.

Reviewed by:

Dr. Rajan Gogna, Ph.D., MS, MBA

Dr. Gogna reviewed this manuscript with help of the PhD working on the area of Cell Competition Ms. Taylor Parker.

Reviewer #1

This reviewer highly evaluates this study and presents some minor suggestions.

1. While the manuscript provide some immunostaining of COX2 and other elements in the pathway, it was not clear to me to which extend does this activation happens. Could the authors provides larger view of the epithelial layer ? Is COX2 upregulation restricted to the neighbourhood of Ras cells or does it extend quite far away ?

According to the reviewer's suggestion, we have shown another image with the larger view in Supplementary Fig. 1b. The data indicate that the upregulation of COX-2 expression can be often observed within two cell-rows from RasV12 cells, which is described in the figure legend of Supplementary Fig. 1b.

2. A previous paper (including some authors of this study, Takanori Chiba et al., Scientific report, 2016) showed that PKC inhibition with BIM1 drugs suppresses Ras cell extrusion (Fig. 3a of this previous paper). This is in contradiction with the results of this new manuscript which suggests that PKC is required for COX2 expression, which inhibits Ras cells extrusion. This may not be so surprising as PKC was shown by the same lab to be required to activate Vimentin, a key component of the Ras cell extrusion (Kajita et al., Nat. Comm., 2014). It would argue again that the same pathway can both activate pro and anti competition components. I would strongly encourage the authors to include and explain this apparent contradiction in the discussion

I sincerely appreciate the reviewer's careful suggestion. As the reviewer suggested, previous studies have demonstrated that PKC inhibitor moderately suppresses apical extrusion of transformed cells. Moreover, PKC- ϵ is accumulated in normal cells surrounding transformed cells, which induces accumulation of vimentin filaments, a positive regulator for apical extrusion. Therefore, it is plausible that PKC phosphorylates multiple positive or negative regulators of cell competition, probably at different stages of apical extrusion; temporal and dynamic regulation of PKC activity can modulate elimination of transformed cells positively or negatively. This issue is discussed from page 10, line 24 to page 11, line 10.

3. At this stage, it is not clear whether this pathway is specific of the oncogenic cell

extrusion. I don't know what is the basal rate of extrusion in homogenous MDCK layer, but it would be interesting to check if PGE2 treatment could also affect normal extrusion.

As the reviewer described, previous studies have demonstrated that high cell density conditions can induce apical extrusion within epithelia. In the revised manuscript, we have shown that PGE2 treatment does not significantly affect the frequency of the crowded cell extrusion (Supplementary Fig. 4b-d), implying that the COX-2-PGE2 pathway suppresses oncogenic cell extrusion rather specifically. This is described in page 7, lines 19-22.

4. The results of this study suggest that inflammation would inhibit cell competition. Most of the data coming from Drosophila rather suggest that pro-inflammatory pathways are required for cell elimination during competition (e.g.: JAK-STAT pathway, work from Erika Bach lab and Eugenia Piddini lab, TNF/JNK, work from Igaki, Moreno and Piddini lab, Toll pathway, work from Johnston lab). I believe this may deserve a little paragraph in the discussion.

According to the reviewer's suggestion, we describe previous studies in *Drosophila* on the role of pro-inflammatory pathways in cell competition from page 11, line 24 to page 12, line 2.

5. How comparable are the intensity of the immunostainings between different cultures (e.g.: Fig. 1e, 2b) ? Are the intensity levels reproducible for a given condition (provided that the acquisition parameters and contrast and maintained) ? This may deserve some explanation in the methods.

For immunostaining analyses, we have captured images using the same set of parameters (e.g. Scan speed/Averaging, Laser power, Sampling Frequency, Pinhole, Detector setting) under each experimental setting to confirm the reproducibility of the obtained results. This is described from page 17, lines 3-6 in the Method section.

Reviewer #2

This reviewer highly evaluates this study and presents several constructive suggestions.

Fig 1: Identification of altered genes in normal cells following co-culture with Ras transformed cells

Authors performed microarray analysis of altered genes in normal cells following (MDCK) cells co-cultured with MDCK cells expressing GFP or GFP-RasV12 and found upregulation of COX2 gene (PTGS2) in normal cells and confirmed upregulation of this gene by qPCR.

No additional experiments are requested.

Fig 2

Pan-PKC inhibitor BIM-1 suppresses the non-cell-autonomous upregulation of COX-2 in normal cells. Results illustrated in this figure suggest that PKC acts upstream of COX-2, however, actual upstream regulators of COX2 are still unclear and some information there will be important to know.

We have further examined the effect of other PKC inhibitors on the expression of COX-2. It has been reported that Ca²⁺-dependent PKC or PKC- ζ can act upstream of COX-2. Go6976, an inhibitor for Ca²⁺-dependent PKC isoforms slightly, but not significantly, suppresses the expression of COX-2 in normal cells surrounding RasV12 cells (Figure a for Reviewer 2; please see below the attached figure). In addition, the PKC- ζ inhibitor ZIP significantly suppresses the expression of COX-2 (Figure b for Reviewer 2). Thus, it is possible that multiple PKC subtypes are involved in the regulation of COX-2.

Previous studies have demonstrated that PKC inhibitor moderately suppresses apical extrusion of transformed cells. Moreover, PKC- ϵ is accumulated in normal cells surrounding transformed cells, which induces accumulation of vimentin filaments, a positive regulator for apical extrusion. Therefore, it is plausible that PKC subtypes phosphorylates multiple positive or negative regulators of cell competition, probably at different stages of apical extrusion; temporal and dynamic regulation of PKC activity can thus modulate elimination of transformed cells positively or negatively. As the reviewer suggested, in future studies, we would like to further examine PKC-catalysed phosphorylation of those cell competition regulators. This issue is described from page 10, line 24 to page 11, line 10.

Fig 3

Cox-2 negatively regulated apical elimination of RasV12 transformed cells by normal cells. Suppression of COX1/COX2 activity increases apical extrusion of RasV12 transformed cells as does surrounding Rasv12 transformed cells with COX2-KO normal cells.

No additional experiments are requested.

Fig 4

Authors used comprehensive quantitative-lipidomics analysis using conditioned media and found significant upregulation of PGE2 in co-culture medium and confirmed PGE2 upregulation using ELISA. The authors next showed that PGE2 secretion negatively regulates apical extrusion of Ras V12 transformed cells in series of reproducible experiments. When PGE2 production was blocked, apical extrusion of Ras V12 cells was rescued.

However, it remains unknown how PGE2 engagement with which EP receptors or how intracellular PGE2 signaling mediates these effects.

Yes, it is indeed the case at present. We have been analyzing the effect of EP4 knockout on apical extrusion of RasV12 cells, but no significant effect has been observed (data not shown). Thus, according to the obtained results so far, it is possible that multiple PGE₂ receptors on normal or RasV12 cells regulate the process of apical extrusion in a concerted manner. Alternatively, not only extracellular PGE₂, but also intracellular PGE₂, might modulate signalling pathways that affect apical extrusion (page 11, lines 21-24). We would like to further elucidate this issue in future studies.

Fig 5: Mouse model of cell competition

Focused on pancreatic epithelia, since Ras is common mutation in pancreatic cancer. COX-2 was elevated in pancreatic epithelia harbouring RasV12-expressing cells as well as other inflammatory markers. Results from in vivo experiments confirmed findings from cell culture models that COX2 negatively regulated apical elimination of transformed cells, but blocking COX2 signaling with NSAIDs enhances elimination of transformed cells.

No additional experiments are requested.

Fig 6

Authors further studied role of inflammation in reducing efficacy of EDAC. Their major finding here was that treatment of mice with caerulein to induce pancreatitis followed by induction of RasV12 transformed cells increased number of RasV12 transformed cells in pancreatic tissue.

Used 3 mice for these experiments, we feel this experiment can have n=5.

According to the reviewer's suggestion, we have analyzed five mice in total. The total number of analyzed ducts are 963 (PBS) and 867 (caerulein) (described in page 29, lines 22-24). The new result is shown in Fig. 6d.

The authors used CD45.2 to prove inflammation, but we suggest other inflammation markers must be included.

A number of previous studies have already demonstrated that the intraperitoneal caerulein treatment that we used in this study induces an infiltration of inflammatory cells into the pancreas leading to pancreatitis (Willemer et al., 1992, European Surgical Research; Lerch et al., 1994, International Journal of Pancreatology; Lugea et al., 2006, Gastroenterology; Omary et al., 2007, Journal of Clinical Investigation). In the revised manuscript, we have performed immunostaining for α -smooth muscle actin (α -SMA), a marker of activated pancreas stellate cells that are accumulated around pancreatic acinar cells upon chronic pancreatitis (Lugea et al., 2006, Gastroenterology; Omary et al., 2007, Journal of Clinical Investigation). As shown in Supplementary Fig. 6a, we demonstrate that caerulein treatment induces the accumulation of α -SMA-positive activated pancreas stellate cells in periacinar spaces.

- MDCK mixed with GFP
- MDCK mixed with RasV12

Figure for Reviewer 2

Effect of PKC inhibitors on the COX-2 mRNA level in normal MDCK cells co-cultured with GFP-expressing MDCK cells (white) or GFP-RasV12-expressing MDCK cells (grey).

a Go6976: Ca^{2+} -dependent PKC inhibitor; **b** ZIP: PKC ζ pseudosubstrate-derived ζ -inhibitory peptide. Data are mean \pm s.d. from five (a) or four (b) independent experiments. * $P=0.022$, ** $P=0.017$ (student' s t -test). Values are expressed as a ratio relative to DMSO or water (MDCK mixed with GFP).

REVIEWERS' COMMENTS:

Reviewer #1 (Remarks to the Author):

The authors have addressed all my concerns. The manuscript is well written and the data very convincing providing very interesting information on the role of inflammation in EDAC both in vivo and in vitro. I therefore fully support publication.

Reviewer #2 (Remarks to the Author):

Fig 1

No additional experiments were suggested.

Fig 2.

Authors have thoroughly explained the multiple mechanisms by which multiple PKC subtypes may be regulating COX2 based on previous literature. Also, authors included an additional PKC isoform-specific inhibitor, ZIP to test potential isoform-specific regulation of COX2. Authors have satisfied initial suggestions.

Fig 3

No additional experiments were suggested.

Fig 4

No additional experiments were suggested.

Fig 5

No additional experiments were suggested.

Fig 6

Comment 1: Authors increased total number of mice from 3 to 5, as requested and strengthened study findings. Authors satisfied initial comments.

Comment 2: Authors included an additional marker of inflammation, alpha-smooth muscle actin, as second line of evidence of inflammation in their mouse model, and have satisfied initial comment.

The authors have provided additional experiments as requested to further validate their findings and have presented consistent results. The authors have also clarified their explanations of their findings, particularly with respect to previously published works. The work described herein adds to the field of cell competition and sheds new light onto the mechanisms by which cell competition fails to protect against cancer initiation in the mammalian epithelium. The work presented is suitable for publication in this Journal.

Dr. Gogna reviewed this manuscript with help of the PhD working on the area of Cell Competition Ms. Taylor Parker.